# Immunoglobulin for Treating Bacterial Infections: One More Mechanism of Action

**DOI:** 10.3390/antib8040052

**Published:** 2019-11-03

**Authors:** Teiji Sawa, Mao Kinoshita, Keita Inoue, Junya Ohara, Kiyoshi Moriyama

**Affiliations:** 1Department of Anesthesiology, School of Medicine, Kyoto Prefectural University of Medicine, Kyoto 602-8566, Japan; mao5615@koto.kpu-m.ac.jp (M.K.); keitaino@koto.kpu-m.ac.jp (K.I.); j-ohara@koto.kpu-m.ac.jp (J.O.); 2Department of Anesthesiology, Kyorin University School of Medicine, Tokyo 181-8611, Japan; mokiyo@ks.kyorin-u.ac.jp

**Keywords:** immunoglobulin, IVIG, LcrV, PcrV, translocation, type III secretory toxin, type III secretion system, V-antigen

## Abstract

The mechanisms underlying the effects of immunoglobulins on bacterial infections are thought to involve bacterial cell lysis via complement activation, phagocytosis via bacterial opsonization, toxin neutralization, and antibody-dependent cell-mediated cytotoxicity. Nevertheless, recent advances in the study of the pathogenicity of Gram-negative bacteria have raised the possibility of an association between immunoglobulin and bacterial toxin secretion. Over time, new toxin secretion systems like the type III secretion system have been discovered in many pathogenic Gram-negative bacteria. With this system, the bacterial toxins are directly injected into the cytoplasm of the target cell through a special secretory apparatus without any exposure to the extracellular environment, and therefore with no opportunity for antibodies to neutralize the toxin. However, antibodies against the V-antigen, which is located on the needle-shaped tip of the bacterial secretion apparatus, can inhibit toxin translocation, thus raising the hope that the toxin may be susceptible to antibody targeting. Because multi-drug resistant bacteria are now prevalent, inhibiting this secretion mechanism is an attractive alternative or adjunctive therapy against lethal bacterial infections. Thus, it is not unreasonable to define the blocking effect of anti-V-antigen antibodies as the fifth mechanism for immunoglobulin action against bacterial infections.

## 1. Introduction

In immunology textbooks, the mechanism involved in the effect of immunoglobulins against infectious diseases is described as having four main features: (1) Complement-associated immunolysis, (2) opsonization (immunophagocytosis), (3) toxin/virus neutralization, and (4) antibody-dependent cell-mediated cytotoxicity (ADCC) (Figure 1) [1,2]. These four mechanisms of action were briefly summarized as follows:(1)Complement-associated immunolysis: After the antibody binds to certain Gram-negative bacteria, spirochetes, or other types of bacteria, the complement components react, pierce the cell membrane, and destroy the bacteria through cell lysis.(2)Opsonin action (immunophagocytosis): Neutrophils and macrophages have a receptor for the Fc portion of IgG and can effectively phagocytose bacteria that are bound to IgG antibodies via this receptor. IgG bound to bacteria induces active oxygen (O_2_^−^) (a bactericidal substance) in the phagosome, which in turn acts directly on the bacterium as an oxidant and promotes phagosome to lysosome fusion. The phagocytosed bacteria are then effectively sterilized and digested by cooperation with lysosomal enzymes.(3)Toxin neutralizing action: An antibody responds by binding to a toxin produced by a bacterium and neutralizing its activity. In the case of a viral infection, the antibody can bind to a virus particle to prevent the virus from entering the target cell.(4)ADCC: When IgG antibodies bind to the virus-related antigens expressed on the surface of a virus-infected cell, a natural killer cell with Fc receptors can then bind to and damage the virus-infected cell. In doing so, it destroys the infected cell, which is the site of virus propagation, and prevents the transmission of pathogenic viruses.

During a bacterial infection, immunoglobulins bind to the surface antigens on bacterial cells to promote bacterial lysis via complement, neutrophil- and macrophage-related phagocytosis, and the neutralization of bacterial toxins [2]. Therefore, the virus-neutralizing action and ADCC are related to the role of intravenous immunoglobulin (IVIG) against viral infections. If limited to bacterial infections, the mechanism of immunoglobulins can be focused on mechanisms (1), (2), and part of (3). In addition to the above-mentioned mechanisms, as a supplementary action, immunoglobulins are thought to bind to the Fc receptors on lymphocytes and suppress cytokine production via stimulation by toxins or cytokines [3,4,5].

From the clinical aspect, nonspecific IVIG is human polyclonal IgG from a pooled plasma of over 10,000 healthy blood donors [6,7]. Because IVIG comes from so many donors, it is thought to contain a wide array of variable regions of antibodies capable of binding to a large variety of antigens [7]. With such an advantage, IVIG has been used to treat severe infections and autoimmune diseases [8,9,10,11]. Because patients with primary immunodeficiency are extremely susceptible to bacterial infection and IVIG can prevent infection in these patients [12], there is no doubt that immunoglobulins in the body have a bioprotective activity that can prevent or cure infections [2]. However, despite the description of the mechanism of action of immunoglobulins in the immunology textbook, with the exception of some infections, such as Group A streptococcus infections [13], no clear clinically significant IVIG effects on severe bacterial infections and sepsis have been shown [14,15]. It has been suggested that this negative result involves multiple factors, such as the IVIG preparation, time of administration, dose, and the inflammatory/immunomodulation profile of the patients [15]. It is highly likely that the heterogeneity of the antibody repertoire for specific toxins and antigens among various IVIG products and product lots has affected the negative results. These situations imply that the selection of target patients and the identification of target antigens for IVIG treatment are more critical for determining whether IVIG therapy can be effective against bacterial infections.

In contrast to the pessimistic aspects of vague clinical effects of IVIG therapy on severe bacterial infections and sepsis so far, important knowledge at the experimental level has accumulated concerning the mechanism of action of immunoglobulins against bacterial infections. Recent advances in our understanding of the virulence mechanism used by Gram-negative bacteria and the host’s immune response specific to the bacterial toxin secretion system (the type III secretion system) raise the possibility of the existence of an unknown mechanism for immunoglobulin-related pathogen targeting [16,17]. Experimental efforts related to these discoveries have suggested the potential for effective antibody therapy. Herein, as a consideration that leads to re-challenging the clinical application of effective immunoglobulin therapy, we overviewed the newly identified toxin-secreting function used by Gram-negative bacteria and the host immunity against it and reconsidered the mechanism of action of immunoglobulins against pathogenic Gram-negative bacteria.

## 2. The Bacterial Type III Secretion System and Its Toxins

In the early 1990s, Swedish bacteriologists reported that in *Yersinia*, the bacterial toxin YopE is directly sent from the bacterial cytosol to the cytoplasm of the target cells through a specialized secretory apparatus [18,19,20]. Later, along with the progress obtained in the genome analysis of many pathogenic Gram-negative bacteria, this new concept of toxin secretion was verified and homologous secretion systems were found in various pathogenic Gram-negative species [21,22,23,24]. This type of toxin secretion, which was then called the type III secretion system, differs from the classical type I and II systems [21]. 

Gene-level analyses have revealed that the type III secretion system evolved from flagella [25,26]. In *Legionella* and *Helicobacter pylori*, toxin proteins and nucleic acids were found to be driven into the target cells by a special secretion device that appears to have evolved from pili. Subsequently, the system was named the type IV secretion system [27]. In both type III and IV secretion systems, toxins are injected directly into the cytoplasm of a target cell using a secretion device [21,22,27]. Therefore, even in the presence of neutralizing antibodies against the toxin, the antibodies cannot neutralize the toxin itself unless the specific antibodies are shuttled into or penetrate into the host cells because the toxin reaches the target cell without exposure to the extracellular environment (Figure 2). The fact that antibody immunization does not enable the host to offset the damage caused by the type III secretory toxins is alarming. Interestingly, toxins are produced in an inactive state in bacteria and only exert enzymatic activity in conjunction with a co-factor after being injected into eukaryotic cells [28,29]. Hence, the inactive toxin only exerts its action after entering the target cell. Thus, although surprising, Gram-negative bacteria are already equipped with a sophisticated stealth intoxication system capable of evading the host’s acquired immunity in the form of antibodies. 

The toxin injection device, or ‘injectisome’, comprises many protein molecules [30]. Toxins are called ‘effectors’, and the mechanism used by the toxins to move from the bacterial cytosol to the outside of the bacterial cell membrane is defined as ‘secretion’ [22,23,24,30]. The bacterium also makes holes in the cell membrane of the target cell to drive toxins into the cytoplasm, a process called ‘translocation’ [31]. Translocation involves a two-protein structure called the ‘translocon’, which is described later [31]. 

## 3. The V-Antigen of the Type III Secretion Apparatus and Its Specific Antibody

In the 1950s, British researchers reported that the protein components secreted from *Yersinia pestis* contain an antigen recognized by the serum from infected mice that exerts a vaccine effect in a mouse model of *Y. pestis* infection [32,33,34,35,36,37]. This antigen was named the V-antigen [32]. Later on, in the 1980s, a set of proteins secreted from *Yersinia* (called Yop, *Yersinia* outer membrane proteins) under low-calcium conditions (named the low calcium-response, *lcr*) were found to contain the V-antigen protein (or LcrV) [38,39]. The *lcrV* gene was then found in the pCD1 plasmid, which is essential for the pathogenicity of *Yersinia* [39], and passive immunity against LcrV was reported [40,41]. In the early 1990s, as mentioned above, it was reported that *Yersinia* injects several Yop proteins directly into its target cells through a special secretion apparatus, and that this apparatus is associated with a set of genes, called the Yop virulon, found in pCD1 [18,19]. Among the five Yop virulon-associated operons, the *lcrGVHyopBD* operon that encodes the five proteins (including LcrV) plays a role in toxin translocation. Key experiments then showed that a knockout mutant of the LcrV gene, *lcrV*, lost its toxicity and that antibodies against it inhibited the toxicity [42,43].

In 1996, the exoenzyme S-related toxins were reported to be *Pseudomonas aeruginosa* type III secretion toxins [44]. In 1997, in addition to two exoenzymes (ExoS and ExoT), the cytotoxic type III secretory toxin, ExoU, was newly discovered as a major lung injury factor in *P. aeruginosa* [45]. Consequently, a region called the exoenzyme S regulon in the chromosomal genome of *P. aeruginosa* was discovered (Figure 3(1)) [46]. Surprisingly, beyond the bacterial species lineage, the exoenzyme S regulon shares high homology with the *Yersinia* Yop virulon [46]. In the exoenzyme S regulon, five operons encode the regulatory proteins, the secretion apparatus, and the translocon components (Figure 3(2)) [46]. The *pcrGVHpopBD* operon, which is homologous to the *Yersinia lcrGVHyopBD* operon, encodes five proteins associated with toxin translocation (Figure 3(2)) [47]. In 1999, the effects of PcrV vaccination and passive immunization with anti-PcrV antibodies in animal models of *P. aeruginosa* pneumonia were reported, as seen with the LcrV vaccinations against *Yersinia,* [48]. In both *Yersinia* and *P. aeruginosa*, specific antibodies against LcrV and PcrV, respectively, reduced the infection pathology [42,48]. 

Translocation of *Yersinia* type III secretion involves two proteins—YopB and YopD—both of which are encoded by the *lcrGVHyopBD* operon [49,50]. The homologs of these *P. aeruginosa* proteins, which are encoded by *pcrGVHpopBD*, are PopB and PopD [43]. YopB and YopD from *Yersinia* [49] and PopB and PopD from *P. aeruginosa* [51,52] are involved in pore formation in the eukaryotic cell membrane. The structural mechanism involving LcrV, YopB, and YopD in *Yersinia* and PcrV, PopB, and PopD in *P. aeruginosa* involves a translocon (Figure 3(3)) [50,52]. The structural position of the V-antigen proteins in the type III secretion system was unknown until 2005, when electron microscopy analysis showed that LcrV and PcrV are both cap-like structures located at the tip of the needle structure in the secretion apparatus [53,54]. Currently, V-antigens are thought to occupy the interface between the secretion needle and translocation as an essential component of the translocon [54].

## 4. Blocking Effects of Antibodies Against the Bacterial Type III Secretion System

The *pcrV*-deletion non-polar mutant of *P. aeruginosa* lost its type III secretion toxicity in the same way as the *lcrV*-deletion non-polar mutant of *Yersinia* [43], but the complementation of *pcrV* with a plasmid in trans restored the toxicity [48]. These observations indicate that a V-antigen, such as LcrV or PcrV, is essential for type III secretion intoxication [42,48]. Active immunization with recombinant PcrV improved mortality in a model of *P. aeruginosa* pulmonary infection [48], and passive immunization with a rabbit-derived anti-PcrV-specific polyclonal IgG against PcrV reduced the acute lung injury associated with *P. aeruginosa* type III secretion [55]. Additionally, intravenous administration of a polyclonal F(ab’)_2_ antibody significantly improved secondary sepsis in a rabbit pneumonia model [55]. The anti-PcrV-specific polyclonal IgG was also effective in a burned skin model of *P. aeruginosa* infection [56] and in a chronic *P. aeruginosa* pneumonia model [57]. Consequently, the mouse-derived monoclonal antibody (mAb) 166 was developed [58], the administration of which reduced lung injury and mortality in the *P. aeruginosa* pneumonia infection model [59]. This effect was also observed with the mAb166 Fab (antigen-binding fragment) [59]. This mAb166 monoclonal antibody was humanized for clinical use as KB001 [60], and phase II clinical studies of the antibody have now been conducted [61,62,63]. Active immunization with recombinant PcrV also improved acute lung injury and mortality in *P. aeruginosa* pneumonia models [64,65,66]. In recent years, more researchers have reported the clinical application of anti-PcrV antibodies, which have been developed using different approaches [67,68,69,70].

## 5. V-Antigen Homologs

To date, type III secretion systems have been found in most pathogenic Gram-negative bacterial species. However, as described above, the target species for detailed studies on V-antigens have been limited to *Yersinia* and *P. aeruginosa* because the existence of an obvious V-antigen homolog has only been found in five bacterial species, namely *Yersinia* LcrV, *P. aeruginosa* PcrV, *Aeromonas* AcrV, *Vibrio* VcrV, and *Photorhabdus* LssV (Figure 4) [71]. It has been reported that rainbow trout vaccinated with recombinant AcrV were not protected against infection with *A. salmonicida* [72,73]. *A. hydrophila* and *Vibrio* are infective to humans, but no detailed studies on AcrV and VcrV from each of them, respectively, have been performed. Thus, with the exception of LcrV and PcrV, the characteristics of V-antigen homologs from other species have not been adequately studied to date. 

As shown in Figure 4, V-antigen orthologs that share a genetic and functional association with V-antigen homologs have also been identified. These include EspA from pathogenic *Escherichia coli*, *Salmonella* SipD and SseB, *Shigella* IpaD, *Burkholderia* BipD, *Bordetella pertussis* Bsp22, and *Chlamydophila* CT1584 (Figure 4) [71]. These orthologs mostly take the form of a coated sheath-like structure around the needle of the type III secretion apparatus, unlike the cap-like structure of the V-antigen homologs on the needle tip [74]. EspA from pathogenic *E. coli* is reportedly essential for type III secretion in this bacterium [75]. Regarding IpaD from *Shigella* [76], although the effects of vaccination with an IpaB–IpaD fusion protein have been reported, there have been no reports on other orthologs to date.

## 6. The Antibody-Blocking Mechanisms against Type III Secretion

Because both the whole IgG molecule and Fab fragment against *P. aeruginosa* PcrV can suppress the intoxication caused by type III secretion, it is possible that the Fc-domain of the antibody is not necessary for the blocking effect [55,59,60]. The binding site of mAb166 is located at position 144–257 of the 294 amino-acid-long PcrV molecule (Figure 5) [58]. Deleting one amino acid from the amino- or carboxyl-terminal of the PcrV fragment at position 144–257 results in the loss of mAb166 binding in immunoblotting, suggesting that this PcrV fragment (144–257) is the minimal blocking epitope for this mAb [58]. In the 3D structures of LcrV and PcrV, there are two coiled-domains in the center and carboxyl-terminal regions, respectively [77,78,79]. These two-coiled regions form a double chain as a coiled-coil center shaft in the V-antigen, and the coiled-coil region forms a dumbbell-like structure with globular domains at both the amino- and carboxyl-terminals (Figure 6(1)) [77,78,79]. In serotype O8 of *Y. enterocolitica*, the carboxyl-terminal globular domain of LcrV contains a hypervariable region at position 225–232, and the antibodies generated against LcrV from this serotype are unable to protect against other *Yersinia* spp. carrying the alternative LcrV type [41]. MAb166 binds to the carboxyl-terminal globular domain of PcrV [46]. Deleting the 144–257 region of the fragment may unravel the coiled-coil state, thereby impairing the 3D protein structure [46,77]. Therefore, the above observations imply that normal mAb166 binding is associated with the conformational structure of PcrV.

The humanized KB001 Fab antibody displays significantly higher binding affinity to PcrV than the original murine mAb166 [60]. KB001 showed significant effects against infections in a dose-response study, suggesting that its inhibitory power against type III intoxication is significantly associated with the binding affinity of this antibody to its cognate antigen [60]. A detailed structure of LcrV has also been reported, revealing a pentagonal ring structure comprising an LcrV pentamer on the tip of the secretion needle [80,81]. Based on the homology between LcrV and PcrV, a 3D structure of PcrV was described for the association with the translocon (Figure 6(2)) [77,78,80,81,82]. Although researchers are not yet certain, antibody binding may occur in the vicinity of the central hole formed by the mushroom-like cap structure comprising the PcrV monomers to physically inhibit the passage of the toxin. These two coil regions form a double chain as a coiled-coil that forms a dumbbell-like structure with globular domains at both the amino- and carboxyl-terminals [77,78,80,81,82].

## 7. The Blocking Antibody Fraction in Human Serum

It has been confirmed that a commercially available IVIG preparation contains a fraction that binds to PcrV, and that administration of the IVIG preparation reduced lung injury and mortality in a mouse model of *P. aeruginosa* pneumonia in a dose-dependent manner [85,86]. This effect was attenuated by removing the PcrV-binding fraction in the IVIG by affinity chromatography depletion [85]. Additionally, a recent epidemiological study showed that anti-PcrV titers in sera from adult volunteers were significantly higher in approximately 10% of the tested volunteers [87]. When the therapeutic effects of the immunoglobulin fractions obtained from the adults with high PcrV titers were tested in an animal model of *P. aeruginosa* pneumonia [88], high anti-PcrV titer-derived IgG was found to significantly improve lung injury and mortality, unlike the immunoglobulin obtained from adult sera with low anti-PcrV titers [88]. These findings indirectly suggest that patients with high anti-PcrV antibody titers have acquired immunity against *P. aeruginosa* type III secretion toxicity. Therefore, clinical administration of immunoglobulin with a high antibody titer is expected to provide a prophylactic or therapeutic effect against *P. aeruginosa* infection.

## 8. The Fifth Mechanism of Action of Immunoglobulin Therapy and Future Research

The above-mentioned findings, which were obtained mainly from *Yersinia* and *P. aeruginosa*, revealed that antibody-binding to V-antigens can counteract bacterial type III intoxication in the host. Crucially in this respect, human sera and human sera-derived IVIG preparations contain antibody fractions that bind to PcrV [85,86,87,88]. The ability of these antibodies to inhibit a bacterial infection is independent of toxin neutralization, complement, or opsonization. Instead, their action seems to physically inhibit toxin secretion and/or translocation by recognizing the 3D structure of the bacterial secretion apparatus. We do not know how significant this immunity is, especially in Gram-negative infections. In the current situation where multi-drug resistant bacteria are common, the application of a V-antigen homolog as a vaccine and immunoglobulin therapy targeting V-antigen homologs is a potentially attractive alternative or adjunctive therapy against lethal bacterial infections. Therefore, it seems reasonable to define the blocking effect of anti-V-antigen antibodies as the fifth mechanism of immunoglobulin action against bacterial infections (Figure 1(5)).

## 9. Conclusions

Four mechanisms of action have been suggested to explain the positive effects of immunoglobulin therapy on bacterial infections. In the type III secretion system, the toxin (an effector) was directly injected into the cytoplasm of the target cell through a special secretory apparatus without being exposed to the extracellular environment, thus providing no opportunity for antibodies to neutralize the toxin. The ability of antibodies to target the V-antigen on the needle-shaped tip of the bacterial secretion apparatus and inhibit toxin translocation suggests that the toxin secretion-inhibitory effects of immunoglobulins should be considered as the immunoglobulin’s fifth mechanism of action against bacteria and as a new therapeutic immunoglobulin strategy. 

## Figures and Tables

**Figure 1 antibodies-08-00052-f001:**
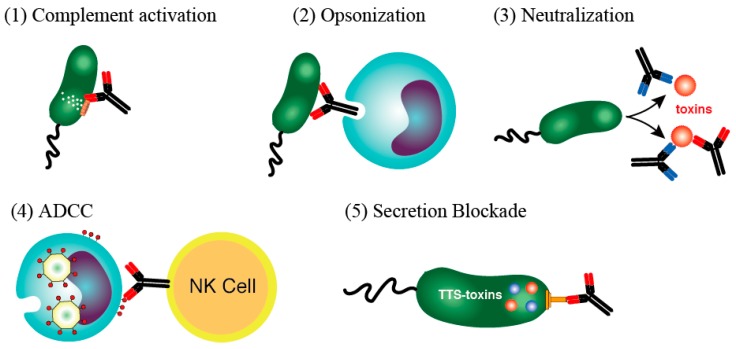
Mechanisms used by immunoglobulins to destroy infectious agents and toxins [1,2]. (**1**) Complement-associated immunolysis, (**2**) opsonization (immunophagocytosis), (**3**) toxin/virus neutralization, and (**4**) antibody-dependent cell-mediated cytotoxicity (ADCC). (**5**) The unmentioned fifth mechanism, which involves the blockade of toxin secretion by immunoglobulins.

**Figure 2 antibodies-08-00052-f002:**
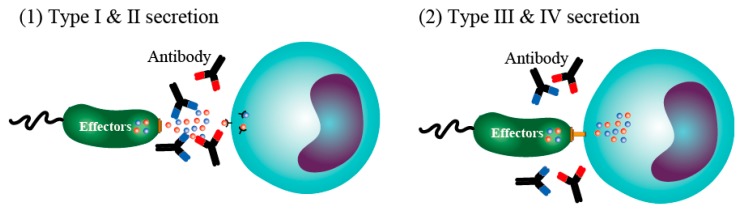
Mechanisms of immunoglobulin-related action against infectious agents and toxins. (**1**) In the classical type I and II secretion systems, bacteria secrete their toxins into the outside environment and immunoglobulins are able to neutralize these toxins. (**2**) In the type III and IV secretion systems, the toxins are injected directly into the cytoplasm of the target cells using a special secretion device.

**Figure 3 antibodies-08-00052-f003:**
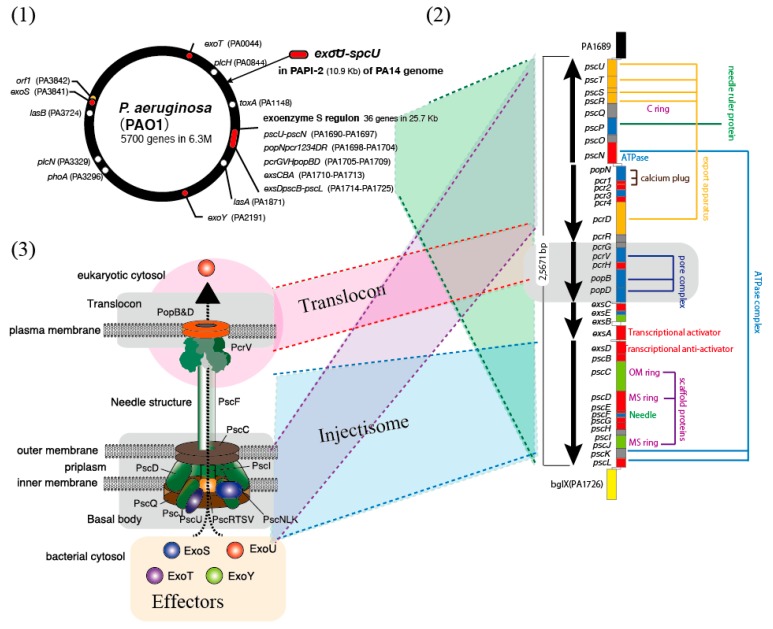
Genome, gene, and protein structures associated with the *Pseudomonas aeruginosa* PcrV V-antigen. (**1**) The *P. aeruginosa* PAO1 chromosomal genome contains a region, the exoenzyme S regulon, which shares high homology with the *Yersinia* Yop virulon [46]. (**2**) In the exoenzyme S regulon, five operons encode regulatory proteins, secretion apparatus, and translocon components. The *pcrGVHpopBD* operon encodes five proteins associated with toxin translocation [46]. (**3**) PcrV is a cap-like structure located at the top of the secretion needle [53], and PopB and PopD are involved in pore formation in the eukaryotic cell membrane [51,52]. The structural mechanism involving three of the PcrV–PopB–PopD proteins involves a translocon.

**Figure 4 antibodies-08-00052-f004:**
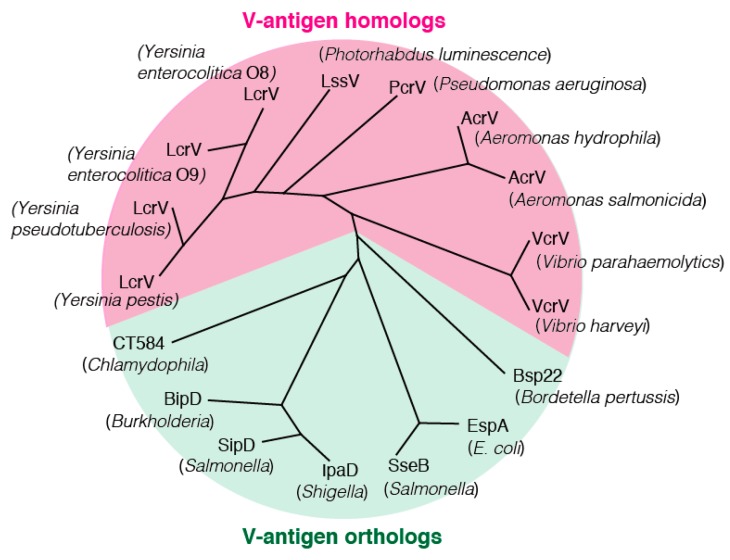
Phylogenetic tree of V-antigen homologs and orthologs. *P. aeruginosa* PcrV, *Aeromonas* AcrV, *Vibrio* VcrV, and *Photorhabdus* LssV are *Yersinia* V-antigen homologs, whereas *Escherichia coli* EspA, *Salmonella* SipD and SseB, *Shigella* IpaD, *Burkholderia* BipD, *Bordetella pertussis* Bsp22, and *Chlamydophila* CT1584 are orthologs of the *Yersinia* LcrV V-antigen [71].

**Figure 5 antibodies-08-00052-f005:**
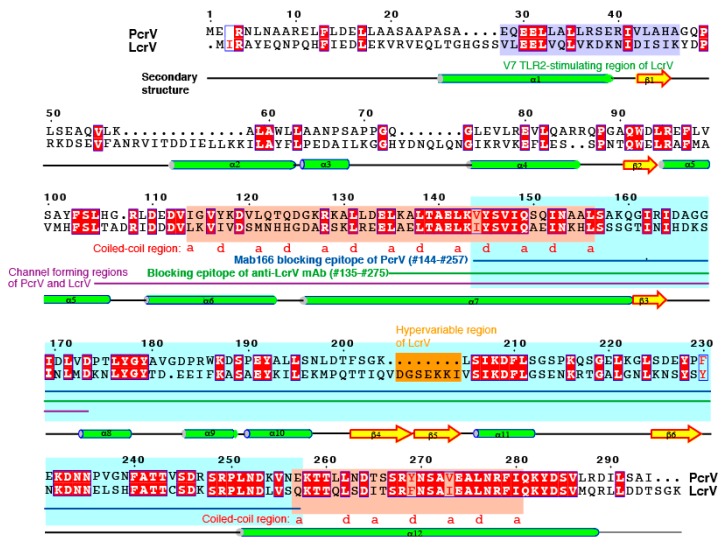
Primary and secondary structures of PcrV and LcrV. *Y. pestis* LcrV is 326 amino acids long and *P. aeruginosa* PcrV is 294 amino acids long. The channel-forming regions of PcrV and LcrV are underlined with a purple line [83]. The blocking epitope of the anti-LcrV monoclonal antibody is located at position 135–275 (a green underline) in the LcrV molecule [38], and the binding site of the blocking mAb166 anti-PcrV antibody is located at amino acid position 144–257 in PcrV (blue underline) [58]. A V7-toll-like receptor-stimulating region at the amino-terminal [84] and a hypervariable region (orange-colored) has been reported for *Yersinia* LcrV [41]. Both LcrV and PcrV contain two coiled domains in their center and carboxyl-terminal regions (designated by the red-colored letters ‘a’ and ‘d’).

**Figure 6 antibodies-08-00052-f006:**
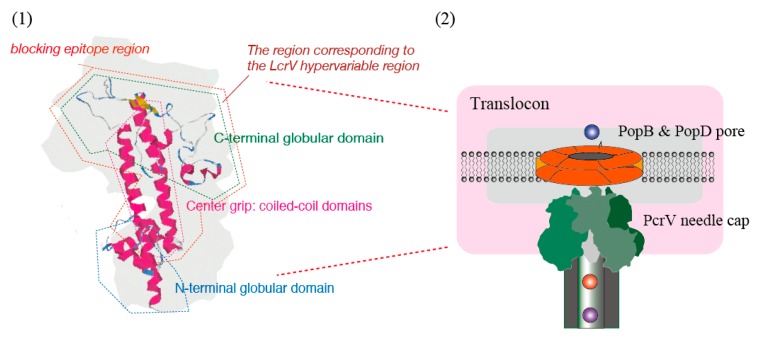
Structures of PcrV and the translocon from *P. aeruginosa.* (**1**) A dumbbell-like structure of PcrV with the central coiled-coil region and globular domains at the amino- and carboxyl-terminals. A hypervariable region has been reported for *Yersinia enterocolitica* LcrV [41]. (**2**) The pentagonal ring structure consists of a PcrV pentamer located on the tip of the secretion needle in the *P. aeruginosa* type III secretion apparatus [77,78,80,81,82].

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
