# Peer review of "Immunoglobulin for Treating Bacterial Infections: One More Mechanism of Action"

_2073-4468, 2019, doi:10.3390/antib8040052_

Round 1

Reviewer 1 Report

In this review article by Sawa et al., the authors present an argument about targeting the type III secretion system (T3SS) of bacteria and more specifically use of antibodies against V-antigen of T3SS to interfere with the delivery of toxins to protect the host from deleterious effects of bacterial infections and to counter the threat of antibiotic resistance in pathogenic bacteria. The manuscript is generally well-written, but consideration of the following points should strengthen the overall presentation.

Further editing of the text is felt to be necessary. Some of the statements/sentences are not quite clear. For example, lines 207-209 and 228-231 on page 7. As shown in Figure 4, Type 3 SS is present in a number of bacterial pathogens, so including the rationale for focusing a major part of the article primarily on Yersinia pestis and Pseudomonas aeruginosa will be helpful. Although this reviewer understands and appreciates that the intent of the article is to project the case for using antibodies to target V antigens, this information is redundant is the text and stated multiple times in the Abstract, Section 1 (Introduction), Figure 1, Section 8 (the fifth mechanism ……. future research), and then conclusions. It would be helpful to remove such redundancy to enhance the flow of the manuscript. For the projected therapeutic mechanism to be effective against intracellular pathogens or toxin delivery systems, the antibodies will have to be shuttled into or able to permeate into host cells. A discussion of this aspect would be welcome.

Author Response

Response to Reviewers’ comments:

#1 Comments and Suggestions for Authors

--------------

Further editing of the text is felt to be necessary. Some of the statements/sentences are not quite clear. For example, lines 207-209 and 228-231 on page 7. As shown in Figure 4, Type 3 SS is present in a number of bacterial pathogens, so including the rationale for focusing a major part of the article primarily on Yersinia pestis and Pseudomonas aeruginosa will be helpful.

Response: We added the following sentences to explain the fact that most of the detailed information about V-antigens is limited inYersinia and P. aeruginosa, even though many pathogenic gram-negative bacteria possess type III secretion systems. 

--------------

Page 6, line 191–195: “To date, type III secretion systems have been found in most pathogenic Gram-negative bacterial species. However, as described above, so far, the target species for detailed studies on V-antigens are limited to Yersinia and P. aeruginosa because the existence of an obvious V-antigen homolog has only been found in five bacterial species including Yersinia LcrV, P. aeruginosa PcrV, Aeromonas AcrV, Vibrio VcrV and Photorhabdus LssV”

Although this reviewer understands and appreciates that the intent of the article is to project the case for using antibodies to target V antigens, this information is redundant is the text and stated multiple times in the Abstract, Section 1 (Introduction), Figure 1, Section 8 (the fifth mechanism ……. future research), and then conclusions. It would be helpful to remove such redundancy to enhance the flow of the manuscript

Response: As the reviewer suggested, we deleted the redundant parts about the basic mechanisms of immunoglobulins in the legend of Fig. 1, and in the Conclusions.  

--------------

For the projected therapeutic mechanism to be effective against intracellular pathogens or toxin delivery systems, the antibodies will have to be shuttled into or able to permeate into host cells. A discussion of this aspect would be welcome. 

Response: According to the reviewer’s suggestion, we added the following segment to the associated sentence.

Page 3, line 108 to Page 4, line 111: “Therefore, even in the presence of neutralizing antibodies against the toxin, because the toxin reaches the target cell without exposure to the extracellular environment, the antibodies cannot neutralize the toxin itself unless the specific antibodies are shuttled into or penetrate into the host cells”.

--------------

Reviewer 2 Report

In the review by Sawa et al, the authors postulate that the beneficial properties of intravenous immunoglobulin for treating bacterial infections is related to an association between γ-globulin and bacterial toxin secretion. The review outlines an overview of the history behind the characterization of type-III toxin secretion systems and provides an overview of supporting literature that indicates that blocking of the type-III system has a potential neutralizing effect. The topic of the review is highly relevant and adds an important element to the discussion of the effects of gamma-globulin therapy. Overall, the review is well formulated and the figures are highly informative and of high-quality. However, the following comments needs to be addressed.

The authors provide a highly optimistic view of the effect of gamma-globulin treatments in the sense that inhibition of toxin secretion is an important mode of action. However, it is this reviewer's opinion that gamma-globulin treatment is predominately used for Group A streptococcus infections and that there is limited or no clinical proof that gamma-globulin treatments is in fact of relevance for gram-negative infections. Because of this unclarities it is important that the authors i) highlight and discuss the failure of providing clinical effect of gamma-globulin for gram-negative infections and ii) add a paragraph that provides a summary of that current status of gamma-globulin treatments in the clinical setting. This is important as lack of clinical efficacy is highly relevant for the claims put forward in this review.

The first paragraph (1. Introduction) is almost completely void of references. The authors should add adequate references to several of the statements in this paragraph. For example, after “With such an advantage, IVIG is used to treat severe infections and autoimmune diseases” and after “During a bacterial infection, IVIG binds to the surface antigens on bacterial cells to promote bacterial lysis via complement, neutrophil- and macrophage-related phagocytosis, and the neutralization of bacterial toxins.”

Additional minor comments:

Is in unclear what the authors mean by “components” in sentence on line 33. Components as in antibodies?: “In fact, the number of components far exceeds the antibody repertoire of a healthy individual [1-5]”

There are a couple of instances where sentences have been duplicated. For example, the following sentence on row 92 appears both in the Figure legend and in the main text. One of these sentences needs to be rephrased: “Therefore, even when a neutralizing antibody against the toxin is present, because the toxin reaches the target cell without being exposed to the extracellular environment there is no opportunity for antibodies to neutralize the toxin itself”

Author Response

#2 Comments and Suggestions for Authors

---------------------

The authors provide a highly optimistic view of the effect of gamma-globulin treatments in the sense that inhibition of toxin secretion is an important mode of action. However, it is this reviewer's opinion that gamma-globulin treatment is predominately used for Group A streptococcus infections and that there is limited or no clinical proof that gamma-globulin treatments is in fact of relevance for gram-negative infections. Because of this unclarities it is important that the authors i) highlight and discuss the failure of providing clinical effect of gamma-globulin for gram-negative infections and ii) add a paragraph that provides a summary of that current status of gamma-globulin treatments in the clinical setting. This is important as lack of clinical efficacy is highly relevant for the claims put forward in this review.

Response: We agree with the reviewer’s point. We added a paragraph that explains the lack of evidence about IVIG therapy for severe bacterial infections and sepsis. In addition, we carefully distinguished the use of the word “IVIG” from the word “immunoglobulins” because IVIG is only one of the immunoglobulin therapies.  

Title: We deleted the word “Intravenous” from the title to generalize the immunoglobulin therapies.

Page 1, line 30 to Page 3, line 82: We rewrote the Introduction to focus on the immunoglobulin’s mechanisms against bacterial infections, instead of IVIG therapy.

Page 2, line 63 to Page 3, line 83: “

From the clinical aspect, non-specific IVIG is human polyclonal IgG from a pooled plasma of over 10,000 healthy blood donors [6,7]. Because IVIG comes from so many donors, it is thought to contain a wide array of variable regions of antibodies capable of binding to a large variety of antigens [7]. With such an advantage, IVIG has been used to treat severe infections and autoimmune diseases [8-10]. Because patients with primary immunodeficiency are extremely susceptible to bacterial infection and IVIG can prevent infection in these patients [12], there is no doubt that immunoglobulins in the body have a bioprotective activity that can prevent or cure infections [2]. However, despite the description of the mechanism of action of immunoglobulins in the immunology textbook, with the exception of some infections, such as Group A streptococcus infections [13], no clear clinically significant IVIG effects on severe bacterial infections and sepsis have been shown [14,15]. It has been suggested that this negative result involves multiple factors such as the IVIG preparation, time of administration, dose and the inflammatory/immunomodulation profile of the patients [15]. It is highly likely that the heterogeneity of the antibody repertoire for specific toxins and antigens among various IVIG products and product lots has affected the negative results. These situations imply that the selection of target patients and the identification of target antigens for IVIG treatment are more critical for determining whether IVIG therapy can be effective against bacterial infections.

In contrast to the pessimistic aspects of vague clinical effects of IVIG therapy on severe bacterial infections and sepsis so far, important knowledge at the experimental level has accumulated concerning the mechanism of action of immunoglobulins against bacterial infections.”

---------------------

The first paragraph (1. Introduction) is almost completely void of references. The authors should add adequate references to several of the statements in this paragraph. For example, after “With such an advantage, IVIG is used to treat severe infections and autoimmune diseases” and after “During a bacterial infection, IVIG binds to the surface antigens on bacterial cells to promote bacterial lysis via complement, neutrophil- and macrophage-related phagocytosis, and the neutralization of bacterial toxins.”

Response: We added the appropriate references to all statements made throughout the manuscript. The number of references increased from 66 to 88, which will help readers refer to extra details. 

---------------------

Additional minor comments:

Is in unclear what the authors mean by “components” in sentence on line 33. Components as in antibodies?: “In fact, the number of components far exceeds the antibody repertoire of a healthy individual [1-5]”

Response: We avoided using the word “component”. Instead, we used more accurate words, such as “array of variable regions of antibodies”, or “the antibody repertoire”.

Page 2, line 65: “Because IVIG comes from so many donors, it is thought to contain a wide array of variable regions of antibodiescapable of binding to a large variety of antigens [2].”

Page 2, line 76: “It is highly likely that the heterogeneity of the antibody repertoire for specific toxins and antigens among various IVIG products and product lots has affected the negative results.”

---------------------

There are a couple of instances where sentences have been duplicated. For example, the following sentence on row 92 appears both in the Figure legend and in the main text. One of these sentences needs to be rephrased: “Therefore, even when a neutralizing antibody against the toxin is present, because the toxin reaches the target cell without being exposed to the extracellular environment there is no opportunity for antibodies to neutralize the toxin itself”

Response: As the reviewers suggested, we eliminated redundant descriptions from the figure legends. We rewrote the conclusions in a simple format. We also changed the figure legends to simpler explanations.

---------------------

Round 2

Reviewer 1 Report

The authors have adequately addressed the comments from the previous review.